# A Threat of Farmers' Suicide and the Opportunity in Organic Farming for Sustainable Agricultural Development in India

**Karthikeyan Mariappan ***[ID] **and Deyi Zhou**

College of Economics and Management, Huazhong Agricultural University, Wuhan 430070, China;
zdy@mail.hzau.edu.cn
*   Correspondence: karthikchachang@gmail.com; Tel.: +86-187-7110-2238

**Abstract:** Agriculture is the main sources of income for humans. Likewise, agriculture is the backbone of the Indian economy. In India, Tamil Nadu regional state has a wide range of possibilities to produce all varieties of organic products due to its diverse agro-climatic condition. This research aimed to identify the economics and efficiency of organic farming, and the possibilities to reduce farmers' suicides in the Tamil Nadu region through the organic agriculture concept. The emphasis was on farmers, producers, researchers, and marketers entering the sustainable economy through organic farming by reducing input cost and high profit in cultivation. A survey was conducted to gather data. One way analysis of variance (ANOVA) has been used to test the hypothesis regards the cost and profit of rice production. The results showed that there was a significant difference in profitability between organic and conventional farming methods. It is very transparent that organic farming is the leading concept of sustainable agricultural development with better organic manures that can improve soil fertility, better yield, less input cost and better return than conventional farming. The study suggests that by reducing the cost of cultivation and get a marginal return through organic farming method to poor and small scale farmers will reduce socio-economic problems such as farmers' suicides in the future of Indian agriculture.

**Keywords:** farmers' suicide; organic agriculture; sustainability in farming; consumer demand; farming practices

---

## 1. Introduction

Agriculture has been proved to be the largest sector in the world economy and also plays a crucial role in the economic development of many nations. However, this sector has been experiencing a lot of crises which need to be dealt with in order to achieve sustainable economic development. Many farmers in various countries such as India, Sri Lanka, Bangladesh, USA, and the UK have been distressed as a result of various factors which leads to suicide on a large scale. The number of farmers' suicides is increasing throughout the world and particularly in India [1]. This problem has called for the formation and implementation of strategies and policies to reduce it. India is observed to be a suicide-prone zone on the continent. In India, many farmers have committed suicide by being overwhelmed by the miseries caused by farming problems such as drought, bad debts, non-production of the crops, and exploitation by private money lenders, poor irrigation facilities, and lack of markets. Many governmental and non-governmental agencies are coming to their aid to minimize the issues of farmers' suicides [2].

Over a long period of time, farmers relied on traditional methods of farming. However, after independence in relation to population growth, the farmers adopt other methods to sustain the

increasing population [2]. New organic methods are being adopted and leading to the production of a high volume of agricultural products [2]. However, despite these improved methods, farmers' suicides have also been increasing due to various factors. The farms are being invaded by insects which are leading to great losses. Due to this series of suicide cases of farmers, the research analyzed the various methods which can be employed to reduce suicide cases to ensure sustainable organic agricultural development [2].

According to Kramer et al. [3], a situation of input dependency of chemical fertilizers and pesticides needs to be erased followed by a move to the remedy of the problem to achieve the sustainability of agriculture. Organic agriculture has opportunities to reduce $CO_2$ emissions by crop management practices [3]. The cases of farmers' suicides in India and Tamil Nadu state have been increasing every year. This has been due to the problems that the farmers are facing in the agricultural sector. Improving organic agriculture in India can reduce these cases of farmers' suicides. The rapid increase in indebtedness is at the root of farmers' taking their lives. Debt is a reflection of a negative economy. Two factors have transformed agriculture from a positive economy into a negative economy for peasants. These factors comprise the rising costs of production and the falling prices of the farm commodities. Both of these reasons are rooted in the policies of trade liberalization and corporate globalization [4].

In order to reduce the issue of farmers' suicides, there are opportunities in organic farming to reduce the input cost and get a premium price for their production. Organic farming helps the farmers to obtain a sustainable household economy. This research is, therefore, analyzing the various scholars and contact an investigation to assess the issues of farmers' suicides in relation to organic and conventional agricultural practices. The cases of farmers' suicides have been as a result of loses that the farmers have been going through due to drought, hence leading to an inability of the farmers to carry out family obligations [5]. The research focuses on the impacts of external sources and activities due to poor agricultural practices and policies in the Tamil Nadu region.

The rate of farmers' suicides in India has been increasing every year due to profit failure in cultivation and other external impacts such as high input cost, drought, climate change, and socio-economic problems. In Tamil Nadu state alone, about 20,000 farmers have died due to poor agricultural reasons up until today [6]. The government and non-governmental agencies have shown their concern towards this tragedy by visiting these areas and offering solutions to the farmers so as to avoid these acts of suicide. An average of 8 farmers commit suicide every day in India [6]. The government of India has been blamed for giving wrong figures of the farmers' suicides in the nation. The proposed reasons for the farmers' suicides in the country are that the rich farmers are able to adapt to the new technology to produce better crops and selling them faster than the poor farmers who only use the manual methods. For this reason, the crops of poor farmers face stiff competition in the market from those of the rich farmers. This condition leads to losses for the poor farmers as they cannot get enough money to pay back the bank loans, hence committing suicide due to huge debts which they cannot settle [7]. The other possible reason for farmers' suicides is that the government does not provide enough training to the farmers who end up investing more on farming, hence using the wrong methods and eventually having poor harvests [8]. This leads to losses to the farmers which is followed by the farmers' inability to pay the available debts and cater to the basic needs of their family and hence commit suicide due to the stress they have. Failure to avail expected amount of credit was quoted as a major cause in all the sample states except Uttar Pradesh and Chhattisgarh [9]. The expectation of institutional credit was highest in Tamil Nadu (80%), whereas the expectation of non-institutional credit was highest in Telangana (68%) [9]. The expectation of loan waiving was cited as a reason for suicide in West Bengal (97%), Kerala (78%), Karnataka (67%) and Tamil Nadu (63%) [9].

Moreover, the government fails to offer support to the farmers, for instance, by providing them with suitable seeds, which enables them to sustain themselves in the rainy reason. The poor farmers end up growing crops which are not drought-resistant, hence making a total loss and suicide. The rate of farmers' suicides is increasing [10] Therefore, there is a need to find a solution towards the problem to save the farmers from this tragedy.

With regard to the aforementioned issues, the study analyzed the cases of farmers' suicide in relation to high input cost of cultivation in Tamil Nadu, India. Also, the significant difference between the type of farming method and the cost of cultivation was examined. Furthermore, this undertaking investigated the significant difference between the type of farming method and the profitability of the cultivation. Therefore, the analysis of farmers' suicide as a threat and the possible opportunities in organic farming methods to reduce input cost and high profit in agriculture for sustainability in Indian agriculture is crucial as it adds to the body of literature. Besides, the findings of this research has vital importance for developing policies so that the farmers of India and other countries with similar situation and problems will be resolved. Moreover, the study highlights the various methods which are necessary for ensuring sustainable organic agricultural development to reduce the cases of farmers' suicides.

## 2. Literature Review

### 2.1. Organic Farming versus Conventional Farming

Organic and conventional farming are the two common approaches to farming in agriculture. These approaches employ different methods and farming practices. On the other hand, both organic farming and conventional agriculture hold different challenges and implications within the global food chain. The agriculture sector is the backbone of economic development in many countries. As Kim [11] asserts, governments have promoted and adopted organic farming for maintaining sustainable agriculture since the last decade. By contrast, farmers commonly practice conventional farming to improve productivity and meet the increasing demand for agricultural products. Kim [11] notes that conventional farming has been an integral part of the agriculture sector since it ensures that it flourished. For this reason, there is an urgent need to address the best options that are available, so that the agricultural sector, not only in India but also in other countries across the globe can be improved [11]. The concept of organic agriculture is a method of efficient use of locally available resources and other agriculture techniques such as fertility management, nutrient cycles, and pest control [12].

Some scholars compare the environmental impacts of organic and conventional farming methods. According to Srendnicka-Tober et al. [13], the global benefits of organic production methods are not limited to reducing soil degradation and erosion but also improving soil structure and fertility, protecting biodiversity, and increasing independence on external production inputs. The aforementioned features of organic production are crucial as far as the protection of natural resources is concerned. From this point of view, the organic farming method supports the concept of sustainable development. On the other hand, conventional farming results in biodiversity loss, environmental pollution, and affects human health among others. It concludes that replacing conventional farming with organic farming translates into measurable economic values.

Organic farming is a system which avoids the use of chemicals in the farms by improving farming methods such as hormones, fertilizers, feed additives, and pesticides. However, the organic farming system relies largely on natural methods such as animal manure, crop residues, crop rotation, off-farm organic waste, use of the biological system of plant protection and nutritional mobilization. Organic farming is, therefore, a unique method of farming which promotes agro-ecosystem health such as soil biological activity, biological cycle, and biodiversity [14]. Organic agriculture is capable of contributing to meaningful socio-economic and ecologically sustainable development, especially in developing countries across the globe. On the other hand, Kilcher [15] asserts that organic farming can contribute to sustainable development with the application of organic principles, which translates into efficient management of local resources like manure and local seed varieties, which in turn, results in cost-effectiveness. In brief, organic farming is capable of contributing to sustainable development since it reduces the risk of yield failure and stabilizes returns, let alone improving the quality of life of small farmers' families [15]. The pros of organic cultivation are such as supporting healthy soil,

pollinators, pest control, being eco-friendly, and offering an opportunity for specializing more nutrition and premium price. The cons of organic farming are that it receives no subsidies, has less infrastructure, there is less demand for organic pesticides, it faces marketing challenges, and there are higher costs at the beginning. While compared with conventional farming the pros and cons are as follows: low production cost, better yield, harmful for people and animals, environment effect, no sustainable economy for small scale farmers. This study chooses to focus on the southern part of India which, specifically Tamil Nadu state. The subject matter will center on productivity as well as the economic status that reflects from the use of both mechanisms of cultivation. Tamil Nadu is well known for its production of rice as it is the largest producer of rice in India, hence playing a pivotal role in assisting the country to achieve self-sufficiency. As stated earlier, governments support organic mechanisms by availing themselves of and facilitating research on the matter. There has been extensive research on the application of organic farming and the problems that are associated with either one or both mechanisms of cultivation [16].

*2.2. Organic Farming in India*

The government of India has tried to promote organic agriculture in the nation at various levels. A lot of money has been put aside in India to assist farmers to improve organic farming. Previous studies conducted showed that various states in India such as Tamil Nadu, Gujarat, Bihar, Madhya Pradesh, and Maharashtra are promoting organic agriculture [17]. Different projects have been initiated in these states to aid in developing organic agriculture. According to the results of the FiBL(Research Institute of Organic Agriculture) 2015, India has about 5.2 million hectares of organic land and organic producers of about 6.5 million [18]. Therefore, India is found to be the largest organic agricultural producers in the world. Additionally, India has the most producers certified through the participatory guarantee system (PGS) followed by Brazil, Peru, and Bolivia. These nations show that there is a possibility of enhancing and promoting organic agriculture as a method of ensuring sustainable agriculture [18].

Organic farming in India has been used since ancient times. It involves a system of farming which is aimed at cultivating the land and producing crops in a way which keeps the soil in good health and alive. Sustainable agriculture is ensured in line with eco-friendly manner through the use of organic waste such as aquatic wastes, farm, crop and animal waste and other biological materials [19]. India is found to possess a lot of potential in organic farming from small-scale farmers to large-scale farmers. Organic farming in India has produced high-quality products to increase the export rates of crops like rice. The agricultural market in India has improved and competition increased as the customers are looking for organic products due to their freshness, micro-organism free and nutrition value characteristics. Organic manure in India is readily available in rural areas making it easier for farmers to use them instead of spending more on commercial fertilizers which have negative effects on the environment. Therefore, the government of India has been emphasizing the use of organic farming methods since chemical fertilizers have led to depletion of soil as well as lowering the quality of products [20].

India is one of the countries which has a large sector of agricultural activities. Some of the products which are a planted in India are rice, tea, herbs, some spices, and others. Before the Green Revolution, farmers used to use traditional ways of farming, in which these practices were environment-friendly. However, after the Green Revolution, farmers adapted the new farming methods whereby chemical fertilizers and pesticide were used, and this leads to a rapid rise in health impacts [21]. The use of these artificial chemicals has also led to pest-resistant which also causing a lot of loss in the agricultural sector. Due to the high expense in acquiring chemical fertilizers which have a negative impact on the soil and environment, farmers in India have started using organic farming. Due to its strong agrarian-based culture, India has continued to promote organic farming by offering marketing services and training. The number of farmers who have adopted organic cultivation has increased at a faster rate. Large areas of land have also been put under organic management in India. This has been as a result of continuous

training of farmers to adopt organic farming discussing with them the benefits of the organic farming practices as well as the disadvantages of the artificial chemicals which have negative impacts on the environment [22]. Table 1 showed the statistics of the land which has been continued to be cultivated under organic farming in India.

**Table 1.** Growth in the area put under organic management in Ha.

| S.NO. | Year | Area Put under Organic Management (in Hectare) |
|---|---|---|
| 1. | 2003–2004 | 43,000 |
| 2. | 2004–2005 | 70,000 |
| 3. | 2005–2006 | 101,000 |
| 4. | 2006–2007 | 123,000 |
| 5. | 2007–2008 | 653,000 |
| 6. | 2008–2009 | 9,090,000 |
| 7. | 2009–2010 | 10,452,000 |
| 8. | 2010–2011 | 12,500,567 |
| 9. | 2011–2012 | 14,098,099 |
| 10. | 2012–2013 | 15,098,000 |
| 11. | 2013–2014 | 14,098,099 |
| 12. | 2014–2015 | 20,000,100 |
| 13. | 2015–2016 | 19,098,009 |
| 14. | 2016–2017 | 21,009,000 |

Source: International Federation of Organic Agriculture Movements (IFOAM 2017).

According to the International Federation of Organic Agriculture Movements (IFOAM), it is clear that the area put under organic management has been growing every year (Table 1). This has been enabled by training efforts for the farmers to adopt organic farming. Large chunks of land which was initially used in artificial agricultural practices have now been turned to organic farming.

Singh et al. [23] had recorded the field experiment and determined that there was no serious pest attack and crop disease by organically grown crops in India. The findings of another study explained that the unit cost of production is lower in cotton and sugarcane cultivation. Meanwhile, for paddy and wheat, the production cost is lower in conventional farming. The cost of cultivation is determined by the success of agriculture [23]. Most of the studies determined that organic farming is more profitable than conventional [24]. However, farmers are reluctant to adopt organic farming. In addition, there is a need to gain more awareness and a clear insight into organic concepts to change the farmers' reluctance to adopt organic farming [24]. Another study in Andhra Pradesh explained that organic agriculture requires a lot of labors compared with conventional farms. In spite of this, the crops in organic farms having different planting and harvesting schedules which may distribute labor demand unequally. It could help the sustainable employment in organic farming [25].

However, Reganold [26] suggests that productivity is not only the main goal but also need to be focused on is the sustainability of agriculture in the long run. Soil fertility and biodiversity maintenance, environmental effects, farmers and their communities are equally important to reach the productivity goals [26].

*2.3. Farmers' Suicides in India*

In conventional farming, there is an urgent need to acquire chemical products such as chemical herbicides and fertilizers. Pavuluri [27] mentioned that the result is that farmers buy chemical products with credits and loans from banks as well as other lenders. Most of the time, farmers are supposed to pay the loan within a period of time and in the event of a default in repaying. However, the farmers are answerable to these lenders. In relation to this, the farmers' day-to-day lives become a struggle whereby they have to cope with debts and the lack of sustainable agricultural produces which would have been the case had they resulted to using organic farming mechanisms. Those farmers overwhelmed by these conditions result in suicide [27]. According to Pavuluri, there is a need for

organic farming but by contrast there is a rapid shift to conventional farming meaning that farmers are now practicing agriculture mainly for the financial benefits rather than for the betterment of health and the ecosystem [27]. Mishra [28] further specifies that the rough estimate of suicides in the agricultural sector in India since the year 2010 can be estimated at approximately 140,000 based on the National Crime Records Bureau (NCRB), which is used as statistics in police records. Mishra additionally stated that NCRB data may be poor quality and inaccurate due to cases of underreporting and the need to cover up the shame or to prevent these cases from being criminal cases. Therefore, Mishra [28] suggests that the number could actually be higher than the stated estimate.

On the same topic, Posani [29] suggests that the main reason why farmers take their own lives is indebtedness and desperation. These main causes were found in all cases of farmers' suicide but, they were just merely symptoms of the problem which is the distress that is growing in the agricultural sector in India. This indicates that there has been a gradual deterioration of the agricultural sector [30]. Other noticeable factors are including high input prices, low profitability, and weak support systems. However, it has been noted that while comparing the suicide rates in different parts of the country, the rate of suicide is high in much more fertile regions as compared to those who live in arid areas where there are fewer resources. This is because the farmer has less ability to deal with shocking events which may include drastic climatic changes or natural calamities [30]. It is also important to note that the rate of suicide may be gradually decreasing but the number of deaths caused by the suicide of farmers is still soaring. In 2016, the rate of suicide of farmers was projected to have decreased by 10%. This puts the figure at 11,370 as compared to 12,602 in 2015 [31]. In Tamil Nadu, the rate of suicides has been dropping since 2014. In another way, the number was 895, 606, and 381 in 2014, 2015 and 2016, respectively. The decrease in rate has been reported in most major regions apart from Karnataka where there was a massive increase from 768 in 2014 to 1569 in 2015 and to 2079 in 2016. Herein, 2016 was the year that recorded the least number of suicides in the past decade [32].

Farmers' suicides are also known as an agrarian crisis. The number of farmers' suicide has been increasing in India. Therefore, there is a need to identify these factors and minimize the number of suicides. Figure 1 showed that the rate of farmers suicide increased by 2% between 2014 and 2015.

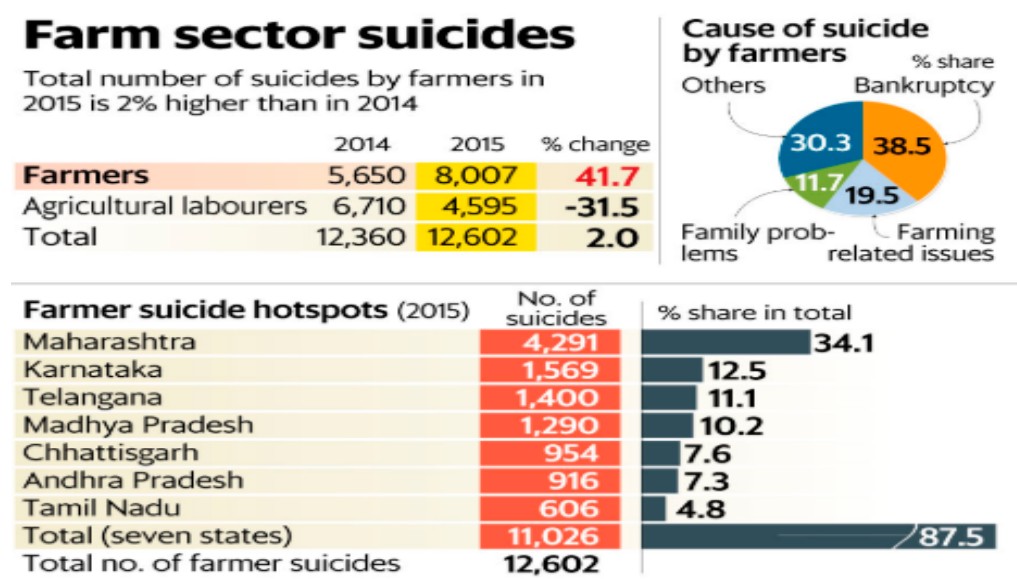

**Figure 1.** The rate of farmers' suicide in 2014 and 2015. Source: National Crime Records Bureau, 2015.

The reasons for the suicides of the farmers is the inability to pay debts taken from the local money-lenders and other institutions [33]. Most farmers are said to drink pesticides to kill themselves. Over 300,000 farmers in India committed suicide between the years 1998–2018. About 70% of the Indian people directly or indirectly depends on agriculture. The percentage of farmers' suicide stands at 11.2% of the total deaths in India.

Scholars and activists have offered a number of reasons as to the phenomenon of farmers' suicides in India. These include high debt burdens, monsoon failure, public mental health, personal issues, family problems, and government policies. Table 2 shows the reasons for farmers' suicide in India.

**Table 2.** Farmers' suicide reasons in India.

| S.NO. | Reasons for Farmers Suicides | Percent (of Suicides) |
|:-----:|:----------------------------:|:---------------------:|
| 1. | Failure of crops | 16.84 |
| 2. | Other reasons (e.g., chit fund) | 15.04 |
| 3. | Family problems with spouse, others | 13.27 |
| 4. | Chronic illness | 9.73 |
| 5. | Marriage of daughters | 5.31 |
| 6. | Political affiliation | 4.42 |
| 7. | Property disputes | 2.65 |
| 8. | Debt burden | 2.65 |
| 9. | Price crash | 2.65 |
| 10. | Borrowing too much (e.g., for house construction) | 2.65 |
| 11. | Losses in non-farm activities | 1.77 |
| 12. | Failure of bore well | 0.88 |

Source: National Crime Records Bureau, 2015.

Agricultural practices in India have declined at a soaring rate since 1990 [34]. The symptoms of unprecedented in post-independence India and agrarian distress shows a high rate of farmers' suicides in India. According to the official government report 2015, 296,438 farmers committed suicide in India from 1995 to 2014. However, information from the agriculturist shows that the number of farmers' suicide is 10 times more than the government's report. The National Crime Records Bureau recorded that in 2014, the number of farmers' suicide in India was 5650. The number of farmers' suicide in India was high in 2004 when 18,241 farmers died due to suicidal actions.

Historical records relating to high mortality rates, revolts and frustration among the Indian farmers, mostly the cash crop farmers, date to the 19th century. The high rate of tax payable to cash crop regardless of famine or drought in that particular year. In Tamil Nadu, the rate of farmers' suicide increases in numbers in 2017.

Approximately 0.5 million farmers have committed suicide in India from 1995 to today. This number has been increasing each year with more cases reported to the Indian states which practice cash crop farming. Table 3 showed the data of the farmers who have been reported to have committed suicide in India from 1995 to 2017. In 1995, the number of farmers' suicide in India was 10,720 and it reached up to 17,130 in 2005. In 2017, the numbers stood at 18,098. This implies that the number of farmers' suicides had increased to an unacceptable level (Table 3).

**Table 3.** The number of farmers' suicide from 1995 to 2017 in India.

| S. NO. | YEAR | No. of Farmers' Suicide in India | S. NO. | YEAR | No. of Farmers' Suicide in India |
|:------:|:----:|:--------------------------------:|:------:|:----:|:--------------------------------:|
| 1. | 1995 | 10,720 | 14. | 2008 | 16,634 |
| 2. | 1996 | 13,730 | 15. | 2009 | 17,345 |
| 3. | 1997 | 13,600 | 16. | 2010 | 17,234 |
| 4. | 1998 | 16,200 | 17. | 2011 | 17,097 |
| 5. | 1999 | 16,100 | 18. | 2012 | 16,990 |
| 6. | 2000 | 16,603 | 19. | 2013 | 17,651 |
| 7. | 2001 | 17,973 | 20. | 2014 | 16,098 |
| 8. | 2002 | 17,809 | 21. | 2015 | 17,045 |
| 9. | 2003 | 17,165 | 22. | 2016 | 16,097 |
| 10. | 2004 | 18,250 | 23. | 2017 | 18,098 |
| 11. | 2005 | 17,130 | | | |
| 12. | 2006 | 17,345 | | | |
| 13. | 2007 | 16,690 | | | |

Source: National Crime Records Bureau, 2015.

The NCRB report [35] indicates that the high number of farmers' suicide have been reported at the Eastern part of Vidarbha which comprises Gondia, Chandrapur, Bhandara, and Nagpur in India. These areas are known for rice farming. Therefore, the cases of suicides are more in the areas where cash crops are grown.

### 2.4. Farmers' Suicides in Tamil Nadu

In Tamil Nadu, agricultural activities are largely carried out. According to the National Crime Records Bureau [35], suicide kills more people in the Tamil Nadu state compared to other forms such as cancer, hepatitis or even HIV/AIDS. The India council for medical research established that cases of farmers' suicides are three times higher in Tamil Nadu than in all southern and western states. Additionally, the researchers have established that the large population of farmers' suicide in the region range between the ages of 20–40 years. The Ministry of Health and Family Welfare also established that the leading causes of premature deaths in Tamil Nadu and other states of India is suicide [35].

Figure 2 showed that the number of farm sector suicides from 2010 to 2015 in Tamil Nadu state. According to the Accidental Deaths and Suicide reports [35], the number of farmers' suicides decreased in 2013 but increased in the year 2014 and 2015 more than in previous years (Figure 2).

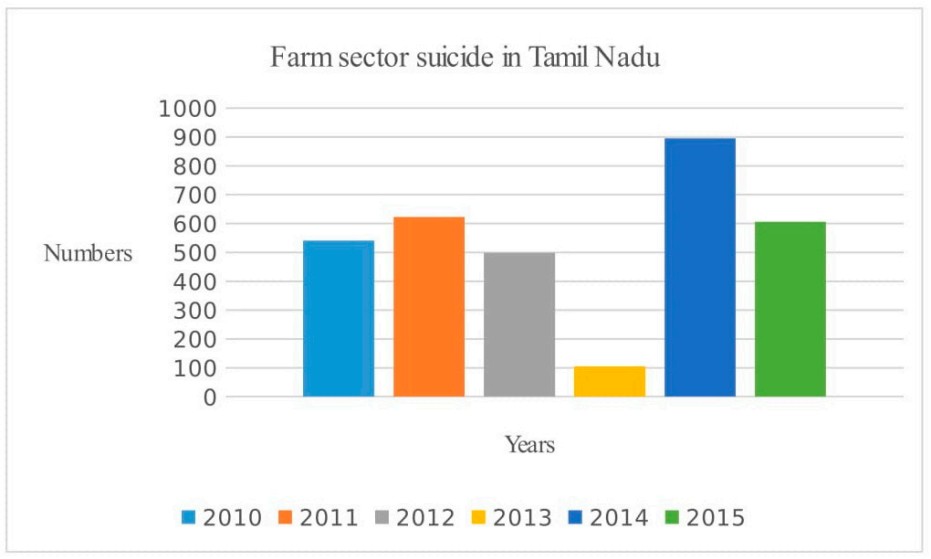

**Figure 2.** Farm sector suicides in Tamil Nadu, 2010–2015. Source: Accidental Deaths and Suicides in India reports 2015.

In 2016 alone, about 25% of the total number of premature deaths were as a result of suicide. The largest number of these deaths were young people aged between 15 and 39 years. Furthermore, the research established that the largest number of victims in Tamil Nadu were farmers. Many farmers are found to commit suicide due to various reasons [36]. First, as a result of their inability to pay debts to one of the leading reasons as to why farmers commit suicide in India. When there are natural calamities like famine and droughts, farmers undergo losses but still the money-lending institution wants their money back. This leads to suicide as they do not have any other source of finance to pay their debts. They feel that it is unnecessary to sell part of their land or look for other methods of paying the debts and end up committing suicide. Conventional farming needs more input cost than organic farming. For this reason, farmers depend on bank and money lenders for their cultivation process [37]. Figure 3 showed that Tamil Nadu is in third place with the most indebted agricultural households leading farmers to endure stress. As we have seen, this has led to detrimental effects like the deaths of farmers due to agricultural distress.

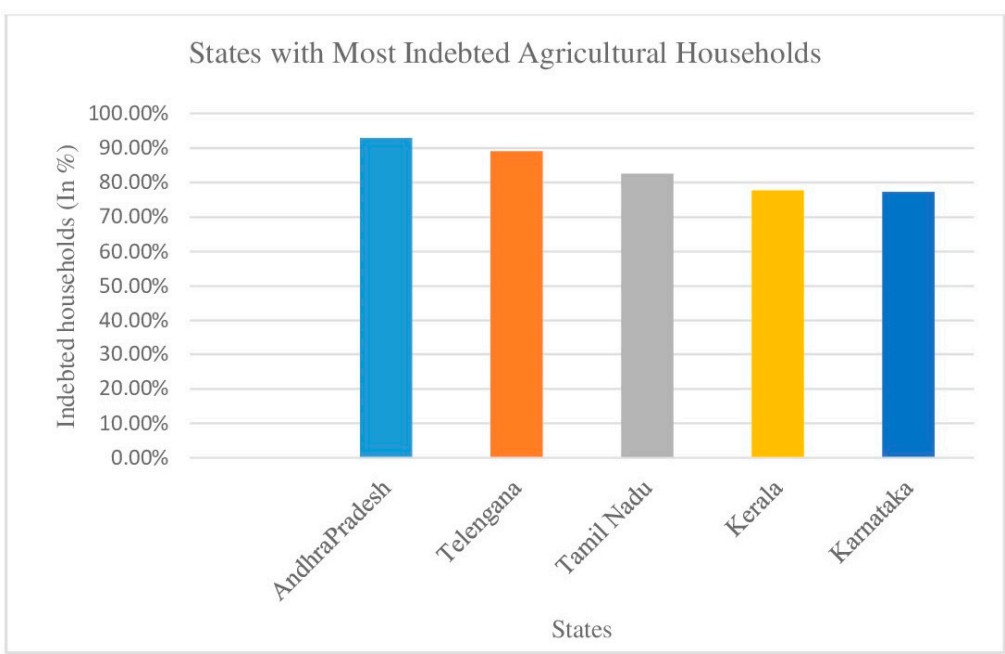

**Figure 3.** States with most indebted agricultural households. Source: Ministry of Statistics and Programme Implementation, 2017.

*2.5. Reducing Farmers' Suicides by Organic Farming*

In India, many agencies have appealed to farmers to embrace organic agriculture due to a large number of suicides. The growth of the market economy in India which doesn't benefit all Indian farmers equally has contributed to increasing the rate of farmers' suicide in India. These markets have been benefited only the large-scale farmers but the small-scale farmers are left with their products unsold or sold at losses. Due to continuous losses, the farmers commit suicide as they cannot cater to their basic needs for their families [38].

Torres et al. [39], concluded that the profitability of organic farming is higher than conventional farming for citrus. The premium price for organic products leads to a high sale price for farmers. These results confirm that organic farming is economically viable and can give guaranteed income through economic sustainability of the households of citrus farmers in Spain [40].

According to Shiva et al. [39], organic farming can reduce the cases of farmers' suicide to a huge extent. Acquiring chemical fertilizers, insecticides and other chemicals is costly. However, embracing organic farming reduces these expenses as it does not involve these synthetic inputs. The farmers need to understand that organic farming makes more profit than others as they do not incur the expenses of purchasing synthetic inputs. Besides, the research has indicated that farmers borrow funds to purchase these synthetic inputs. When there are droughts, famines or poor markets, the farmers who had borrowed funds from the money lending institutions find it difficult to pay the funds due to losses. This contributes to suicide to many farmers. It is, therefore, important for the farmers in India and around the world to embrace the utilization of organic agriculture so as to reduce these cases of suicides [39].

Another study from the consumer side explained that organic farming is a new method to avoid the costs which have been associated with conventional farming and it is considered to have ecology and environment benefits as a whole. Still, the benefits of organic farming are not well known, it will take more time and effort to make organic farming to reach in the center for farmers and consumers [41].

### 2.6. Consumers' Demand for Organic Farm Products

Consumers have increased their demand for organic products at a soaring rate. The consumers are claiming that organic farm products are much healthier than artificial ones. According to the research, artificially developed farm products cause a lot of health problems to the consumers and, therefore, consumers are adopting farm products that have been developed through an organic method for health reasons. This increase in demand for organic farm product has benefited the farmers a lot who are now continuing to improve their farming by expanding their land for organic agricultural practices. The bar chart below shows the increase in demand for organic products by consumers from 2004 to 2017 [42]. Figure 4 explains that the huge demand for organic products in the global market. This will lead the farmers to adopt organic farming for the market opportunity and the premium price for organic products. The chart indicates that the demand has risen from 20 billion to 120 billion US dollars from 2004 to 2017. The data can motivate farmers from reluctance to hope to convert to organic farming (Figure 4).

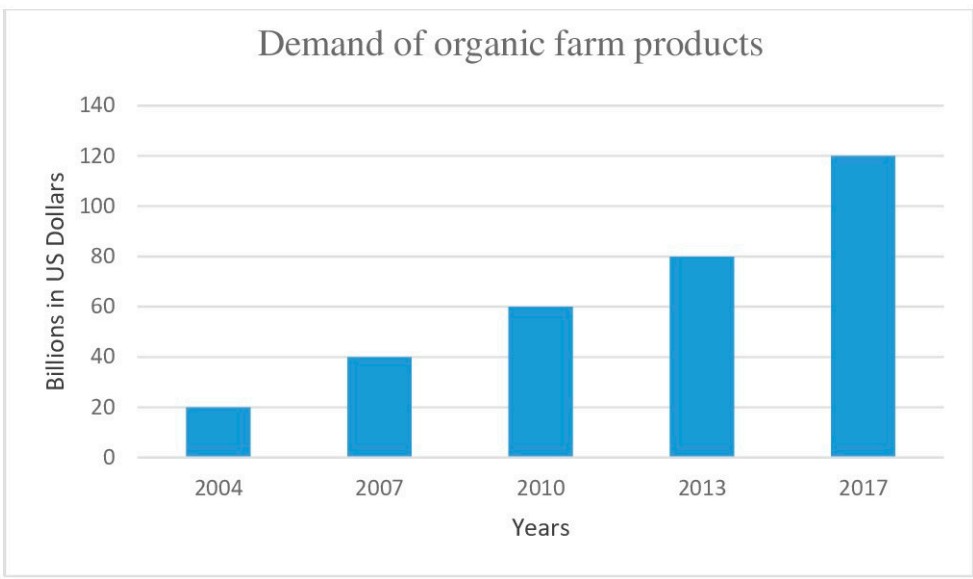

**Figure 4.** Demand for organic farm products. Source: Agricultural and Processed *Food* Products Export Development Authority (*APEDA*).

## 3. Research Methodology

In collecting primary data, both general and specific information was used. The general information includes the gender of the farmers interviewed, age, family type, family size, type of farming mechanism used, village, district, contact details, level of education and farming experience. The specific information used included cropping production cost information like total production in kilograms, the amount of productivity used for self-consumption as well as those used for commercial purposes and the total revenue earned by each farmer. Land use in Tamil Nadu is in M ha. Total cultivable area is 8.16, irrigated area 2.80%, or 34% rain fed, dry land 5.36% or 66%. This research size in the Dryland agricultural area of Tamil Nadu which has 66% of the cultivable area. Among 32 districts of Tamil Nadu, Pudukottai, Madurai, and Sivagangai districts have been choosing as they are predominantly cultivating the selected crop (paddy) in dryland agricultural area. The state of Tamil Nadu is the study area and mainly focused on the food crop; paddy is the selected base for the proportion of the cultivation. This study is based on both primary and secondary data will be collected from various sources. A pre-tested and well-designed schedule has been canvassed among the selected sample holdings to elicit information on the structure of farm holdings, demographic characteristics, asset structure, cost of cultivation, returns, etc. As this is quantitative research, 200 paddy-cultivating

households have been selected comprising of 100 organic farmers and 100 conventional farmers' households from Pudukottai, Madurai, and Sivagangai districts of Tamil Nadu. The field study was carried out and the data had been collected from the sample farmers is related to the cropping year 2016–2017. The raw data collected in this study was condensed, grouped and summarized in pie charts, tables and bar charts. To test the significant difference between the type of farming method and the cost of cultivation. These results lead to adoption of the less input cost and highly profitable farming method by poor and marginal farmers. Through this scenario, they will have a stable and sustainable profit by the organic farming method and this helps to reduce the farmers' suicides for agricultural reasons. A one way analysis of variance (ANOVA) test has been used to check the hypothesis and also analysis the cost and productive efficiency of both farming method. The comparative studies between organic and conventional farms, efficiency analysis is particularly suitable for assessing the farmers' relative ability in optimizing internal resources [43].

The nature of the research is divided into four different variables. The first one is the dependent variables. This can also be referred to as the net income or the profit. In the study, the net income or profit that we would get would be the dependent variable. The second variable is the independent variable. These are the aspects that affect the dependent variables. While interviewing farmers, the questions are intended to compel answers that would show the various factors that would affect the incomes and profits in whichever mechanism adopted for cultivation. These variables include the geographic position of the area of study and the input cost. Input costs include those spent on farming methods, yields or market prices. A comparison is normally made between the input cost and the yield. The third variable is the yield or productivity. This is crucial, as the choice of mechanisms to use will determine the amount of produce from an agricultural project. Other factors come into play like the type of manure used, be it organic manure in the form of animal waste or chemical fertilizers and herbicides. As discussed earlier, this is also a choice of the type of farming methods used and would have a direct effect on the rate and quality of products. Lastly, the fourth variable is market prices. This final variable is important as it is directly used in the calculation of profits and income, hence it would assist in determining whether the agricultural project sustains the farmer financially or whether they would be undergoing constant losses. As mentioned earlier, financial gain or losses may ultimately lead to grave outcomes like suicide. Therefore, market prices are important as they are directly compared to the cost of cultivation.

Additionally, most of the land is dry and dependent on external facilities of irrigation instead of natural rain. This also contributes to the over-reliance on external factors for cultivation hence comparable to conventional farming. This is derived from the field study and is expressed below (Table 4).

**Table 4.** The frequency level of the land type which has been used in the region.

|  | Land Type | Frequency | Percent | Valid Percent | Cumulative Percent |
|---|---|---|---|---|---|
| Valid | Both | 9 | 8.6 | 8.6 | 8.6 |
|  | Irrigated | 85 | 81.0 | 81.0 | 89.5 |
|  | Rain fed | 11 | 10.5 | 10.5 | 100 |
|  | Total | 105 | 100 | 100 | 100 |

Source: Primary Survey conducted during 2016–2017 cultivation year.

## 4. Results

Figure 5 showed that the crops are grown in the study area which is in Tamil Nadu region, India. This research focused on the commercial crop rice which has the maximum proportion of the area under cultivation. Apart from paddy, banana, cotton, hybrid rice varieties, pulses, sugarcane are also familiar crops grown in the region of Tamil Nadu (Figure 5). As deduced, traditional rice is the most popular crop cultivated in Tamil Nadu which is mostly cultivated by the organic farming method.

The total production amount of rice is approximately 35.2%, followed by groundnuts which are about 25.7% of the total (Figure 5).

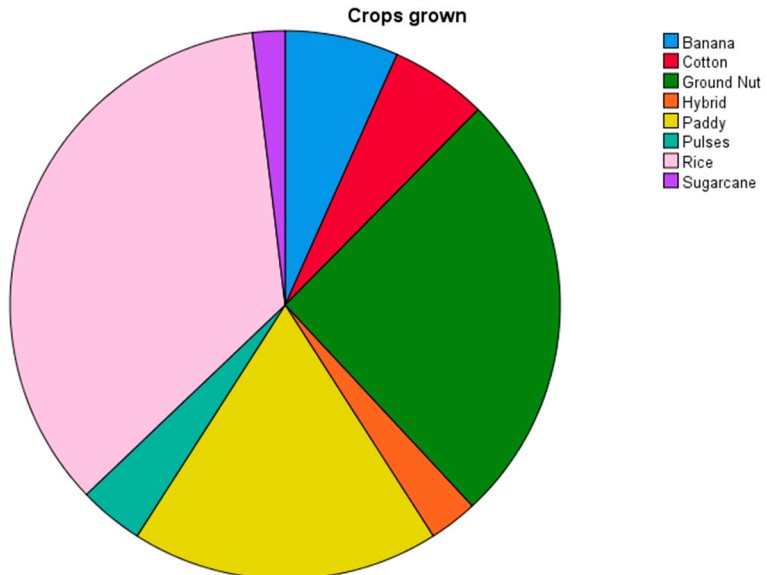

**Figure 5.** The estimation of cultivated crops in Tamil Nadu region. Source: primary survey conducted during 2016–2017 cultivation year.

The second point of analysis was a comparison between the input cost and revenue for both farming mechanisms (Figure 6). The data used to determine the input cost include the total costs used for labor, chemical or organic fertilizer used depending on the preference of the farmers and the cost of machinery. Generally, viewing the results, there is no much difference between the average input costs of the mechanisms. Similar labor and machinery costs were observed and this may be attributed to the similarity of the average input costs between the different methods.

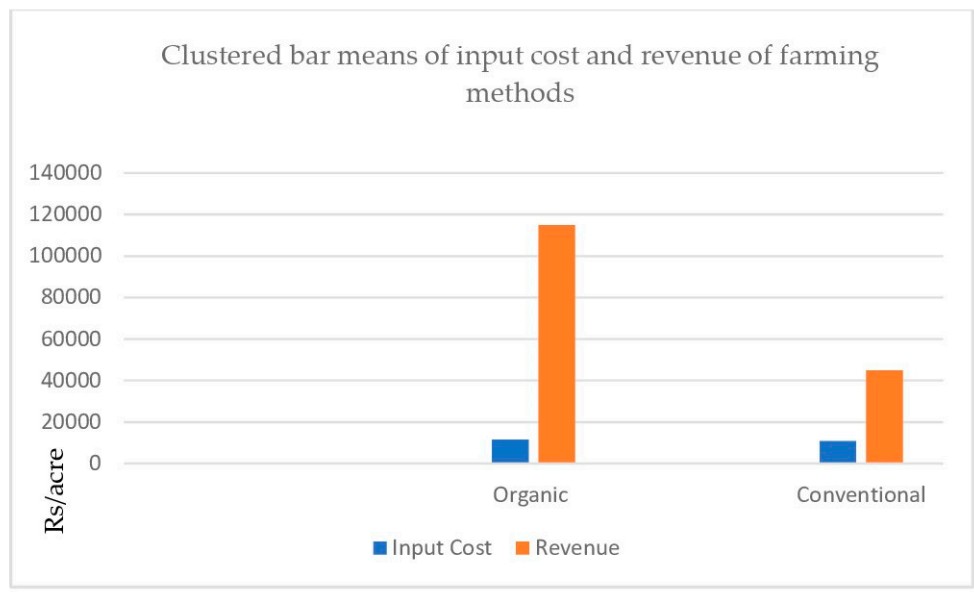

**Figure 6.** Clustered bar means of input cost and revenue of farming methods. Source: primary survey conducted during 2016–2017 cultivation year.

As per the results, the input cost of both farming methods is not very different (Figure 6). Figure 6 indicates that the input cost in organic farming was 11,600 INR (Indian Rupees) per acre. Meanwhile,

the input cost in conventional farming was 10,800 INR (Figure 6). Due to the demand for labor and the increase of machinery cost there is an imbalance of input cost in organic farming.

Lastly, one of the other major areas of analysis were market forces (Table 5). Different farmers have different markets for their harvested produces. Most organic farmers supply their produce to the local marketplace (Table 5). Most of the farmers practicing organic farming have their markets at local villages whereas conventional farming is attached to external markets. This allows the farmers to be flexible with their prices in order to facilitate accountability of the total input cost of cultivation as well any financial gain received from the agricultural practice. Urban markets are the second major source for the organic farmers to sell their products in semi-urban, urban, and global market (Table 5). They have collected the agricultural products from farmers through their networks and sales in urban and global markets. This helps the farmers to obtain more profit for their products and sell their products in the international market (Table 5). The market for conventional farmers includes customer-based markets as well as corporate entities (Table 5). The market is, therefore, an important variable as it may directly affect the financial gain and the relationship between the cost of investment and the percentage of returns.

**Table 5.** The frequency level of markets used by farmers.

| Market Type | Frequency | Valid Percentage |
|---|---|---|
| Urban Market | 19 | 18.1 |
| Customer Base | 13 | 12.4 |
| Village Market | 73 | 69.5 |

Source: Primary Survey conducted during 2016–2017 cultivation year.

### 4.1. Cost and Productive Efficiency (Profit) of Organic and Conventional Farming Methods

The cost and productive efficiency of organic and conventional farming methods were analyzed using one-way analysis of variance (Table 6). The result showed that the input cost per acre mean comparison between organic and conventional farming methods was not significant at *p*-value 0.05. This implied that input cost per acre for both farming methods did not have a significant difference. On the other hand, a profit means that a comparison difference between organic and conventional farming methods was significant. These findings showed that the profit of the farmers in both farming methods has a significant difference. Accordingly, the H1 is rejected and H2 is accepted (Table 7).

**Hypothesis 1 (H1).** *There is no significant difference in the input cost per acre between farming methods.*

**Hypothesis 2 (H2).** *There is no significant difference in profit per acre between farming methods.*

**Table 6.** Mean comparison of cost and productive efficiency (profit).

| | | Sum of Squares | df | Mean Square | F | Sig. |
|---|---|---|---|---|---|---|
| Cost per acre | Between Groups | 13,918,358.740 | 1 | 13,918,358.740 | 0.180 | 0.672 |
| | Within Groups | 7,976,507,703.631 | 103 | 77,441,822.365 | | |
| | Total | 7,990,426,062.371 | 104 | | | |
| Profit per acre | Between Groups | 117,072,471,736.468 | 1 | 117,072,471,736.468 | 18.032 | 0.000 |
| | Within Groups | 668,713,513,411.466 | 103 | 6,492,364,207.878 | | |
| | Total | 785,785,985,147.935 | 104 | | | |

**Table 7.** Hypothesis tests.

| Variables | Sig. Value | Results | Inference |
|---|---|---|---|
| Cost of input per acre | 0.672 | H1 is Rejected | There is no difference in the cost incurred between conventional and organic farming methods. |
| Profit per acre | 0.000 | H2 is Accepted | There exists a difference in profit (productive efficiency) between conventional and organic farming methods. |

The cost and profit mean of conventional and organic farming methods is depicted in Figure 6. The result showed that the mean of the cost of the conventional and organic farming method was, 10,800 INR in conventional farming and 11,600 INR in organic farming, respectively (Figures 6 and 7). The profit mean of the farmers was 45,000 INR in conventional and 115,000 INR in the organic farming method, respectively (Figures 6 and 7). The result implied that there was not much difference in the cost of agriculture observed among organic and conventional farming methods. But there was a difference in profitability between organic and conventional farming methods.

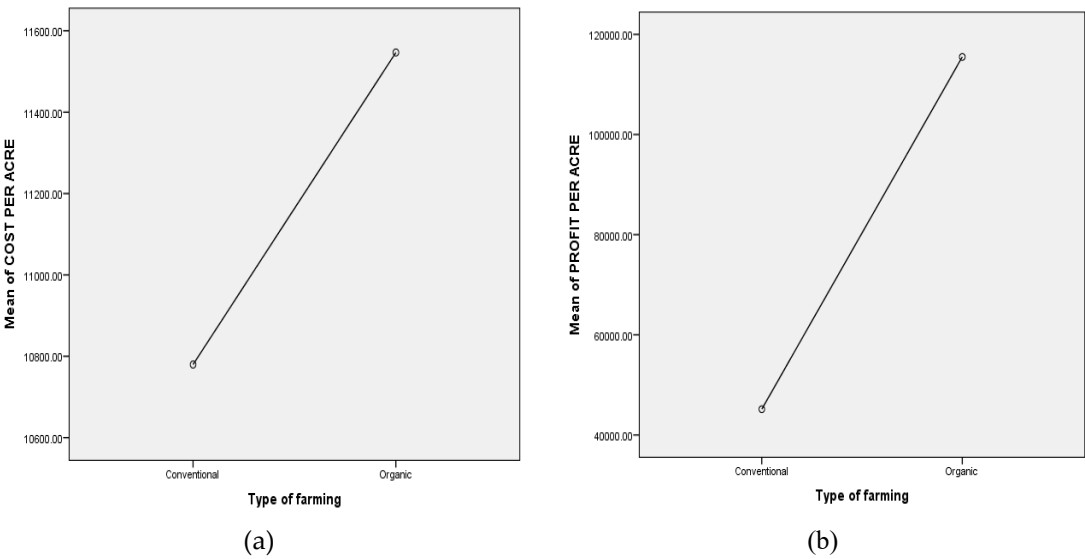

**Figure 7.** Cost and profit mean for conventional and organic farming methods. (**a**) The cost mean of farming methods; (**b**) The profit mean of farming methods.

## 4.2. Discussions

After the analysis and presentation of the data, it is clear that the rate of farmers' suicides has been increasing every year in India. Therefore, there is a need for a serious action to be taken to prevent this number from rising. There is a need for the government to engage with farmers through face to face conversations in seminars and organized events. If the government could go to local areas where the farmers are, then they could be in a good position to offer solutions to farmers, hence avoiding further suicides. The government should also start offering deep training to the farmers about the new methods of farming which could help them increase the quality and quantity of their products. Additionally, the government should look for the market for the cash products of their farmers which could help them to avoid poor prices which are leading to losses and then suicides. If the government could take responsibility for increasing awareness among the farmers concerning the right farming methods, then the rate of farmers' suicides can be reduced to a great extent. In this study, the input cost of both farming methods did not have much difference, even though the profitability through different market channels and the premium price of organic products, the experienced farm, and farmers of the organic farming method and the global demand of organic food determined that the net profit in organic farming is higher than conventional. These results help the farmers, agriculturists, and policymakers to adopt organic farming as a leading agricultural method for sustainability in the Indian agricultural sector.

Apart from this, the discussion part also describes the unique and difference of this research results with the findings of other studies. As the results here showed that the significant difference in the profitability of organic farming. Previous studies' findings on the profit of organic and conventional farming methods e.g., Chandrashekar, H.M. (2010) [2], Mishra, B.B. (2004) [6], Nandwani (2016) [7], Forster, D. (2013) [12], Charyulu, D.K. (2010) [14], Singh, Y.V. (2007) [23], Sudheer, P.S.K. (2013) [25]

determined that the profit in organic farming is higher than that of conventional farming, which is similar to the results of the study at hand. Meanwhile, comparing with input cost the results of this study shown that there was no significant difference in both farming methods. Results related to input cost differ from previously studied findings. The reasons behind this are that the current organic agricultural situation of the study area has an impact on the results through state and central government policies, labor shortages, fewer subsidies, an increase of marketing cost, lack of awareness, lack of infrastructure, and a shortage of husbandry. On the other hand, the Indian government statistical report emphasises the agrarian reasons for suicide which fully depend on the profit and sustainable income of the cultivation. The opportunities in organic farming for economic and environmental benefit are higher than conventional. These study results accentuate the higher profitability of organic farming as a way to prevent suicide in the future.

All the findings of other studies similarly show that the cost of cultivation and profitability in organic farming is relatively different from conventional farming. In other aspects such as yield, labor, the market price is also crucial for policy makers and farmers to have deep insight about organic farming. Even the input cost of organic farming for paddy cultivation in Tamil Nadu also has similarity in results but have a significant difference in profitability than conventional. It leads to the importance of the organic farming benefits in economic and environmental aspects to reduce the cost as much as possible to have a better and sustainable economy. Organic farming concepts and methods help to household's economic sustainability to reduce the farmers' suicide by the causes of agrarian reasons.

## 5. Conclusions

In conclusion, farmers' contribution to the economy through organic cultivation and it is a pivotal step towards the improvement of the quality of farm products and healthy soil which will ultimately contribute to a healthy environment. A healthy environment means a healthy livelihood. On the other hand, despite the popularity of conventional farming, it has been established that the mechanism is not cheap and those who have made agriculture a way of life become frustrated by the external pressure to adapt and cope with the expenses that accompany conventional farming. The government should not only encourage consumers but also farmers to adopt the organic farming mechanism. This can be so by the creation of separate marketing channels for organic produces, creating demand by encouraging more awareness programs, making announcements on premium prices for staple foods, investment in organic farming, and cheap and quick certification processes.

It is also important to note that organic farming practices have a great impact as far as sustainable development is concerned. Organic farming can be used to reduce these cases of farmers' suicides in India and other parts of the world. Most of the cases of farmers' suicides have been as a result of farmer's inability to settle debts which they have incurred while acquiring artificial chemicals. Additionally, farmers should be aware that artificial agricultural methods have negative impacts on the environment and hence should be avoided. Organic farming methods ensure that there is sustainability in the agricultural sector due to its environmental and economic aspects.

**Author Contributions:** Conceptualization, K.M. and Z.D.; methodology, K.M.; software, K.M.; validation, Z.D., and K.M.; formal analysis, K.M.; investigation, K.M.; resources, K.M.; data curation, K.M.; writing—original draft preparation, K.M.; writing—review and editing, K.M.; visualization, Z.D.; supervision, Z.D.; project administration, Z.D.; funding acquisition, Z.D.

**Funding:** This research was funded by the project "the policy impacts of introducing green electricity quota trading system on the sustainable electricity development in China", Project No: 13YJA790163 and The APC was funded by the project No: 13YJA790163.

**Conflicts of Interest:** The authors declare no conflict of interest.

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
