# Peer review of "A Threat of Farmers’ Suicide and the Opportunity in Organic Farming for Sustainable Agricultural Development in India"

_sustainability, doi:10.3390/su11082400_

Round 1
Reviewer 1 Report
Dear Authors, thank you for submitting your manuscript. I find the topic of your manuscript really important and potentially very beneficial. I am glad you are doing your research on this interesting topic.
However, in the present form, your manuscript suffers from some serious issues that need to be addressed.
Generally speaking, there is far too much repetition of the same things and concepts over various sections, and far too little focus on the “red line”. In my opinion, the literature review should be integrated into the introduction part. For now, the manuscript seems to address mainly governmental statistics and some conceptual works. You should include more scientific articles and up-to-date research in the introduction, and compare your findings to these findings in a clear way in the discussion. The methods and the results should be presented in a clearer way, with clearer messages and results. There is no place for repetition of the concepts in the methods section. Also, stay clear of generalized statements and common knowledge. Again, the “red line” needs to be defined and followed throughout the manuscript. Provide evidence for everything you write in the form of references, that is the scientific path. Please have your manuscript proofread by an English speaker. There are a lot of grammatical and syntax mistakes that need to be corrected. Just as a small example; you are writing about farmers’ suicide (plural) not suicide of a single farmer (farmer’s suicide). The sentences are often quite repetitive, oversimplified in their structure, and need to be worked on.
Here are some more particular suggestions:
The Title needs more clarity. You are actually not comparing farmers’ suicide with organic farming, but from what I can deduce, Farmer’s suicide of organic vs. conventional farming.
The Abstract
17-19 I do not understand what you want to say. Please say it in a simpler way.
Introduction:
45-54 Some of the common terms do not have to be defined. For example, the definition of suicide is not really needed.
66-67 You write that “According to research, it is clear that poor organic agricultural practices have a powerful correlation to the farmer’s suicide.” However, you provide no references for this statement. Please provide some references that confirm this.
76-77 I take it that you want to analyze the relationship between farmers’ suicide rate and the use of sustainable agricultural practices? You should also define in the background of the study what you consider here to be sustainable agricultural practices and why.
75-84 Please just write about the main objectives briefly. The current structure is difficult to read. Also, no need to have subsection 1.3 and 1.4. In fact, you don’t need any subsection in the introduction.
85-93 Some significance statements seem repetitive. Please only write about the most important outcomes.
In general, the Introduction section has almost no references to any research done so far. This point needs to be corrected. Briefly include some main studies that have investigated suicide rates of Indian farmers so far and possibly also looked into the farming practices that were being used. These studies can then be further expanded in section 2.
Section 2
This section is quite broad, lengthy, repetitive and general. I would suggest building a clearer “red line” in the literature review. And perhaps including the review in the introduction part. In any case, here you need to include even more studies, not just pure governmental statistics. Avoid repeating the same topic over more than one sub-section. You go into descriptions a lot, e.g. describing the location of Tamil Nadu, Organic Farming Concepts and Principles… In my opinion, these are not really necessary and the information can be condensed a lot here. At the same time, try to introduce more scientific studies. Do not explain them at length, but keep to the point- what were their findings.
95-135 This part is very general, in my opinion too broad and provides no references for the statements. Is this part even necessary? If yes, please condense it.
144-146 Please provide a reference for your statement.
150 reference 12, please only quote the last name of the author, no need to put Dr in front of his name. Also, please enumerate all your references by the order of appearance in the text. 1,2,3 etc…
153 Please quote the author together with the reference number
180 Please put the reference number.
Section 3
I would like to have some more clarity on your sample. The number of farms, where they are located, etc… Here you say you mixed qualitative and quantitative research. What were the outcomes of qualitative research? How was it conducted? Even if you mention all of this, it needs to be presented in a clearer way, otherwise, the reader gets lost.
I recommend splitting the methods and the results in separate sections with more clarity.
358, 361-362 Please avoid stating the obvious.
417-422 This paragraph does not belong to the Methods. It is a sort of repetition from what was already said in Section 2. I would suggest integrating it there.
426-432 Please provide some sort of evidence for this statement. Is it a finding of someone else or your own result?
464-503 Again, a repetition of some concepts already mentioned in section 2. This part needs to be either removed or integrated into the literature review.
Image 3.3. Very generalized concept. Is it required in this manuscript? Does it serve the red line in any way? Neither a method nor a result.
Discussion
518-523 Again repetitive. Please bring the general statistics to Section 1 or 2 and just discuss them here. The discussion is not meant to present any new statistics, just compare and discuss your results with those of others. This part is missing.
Author Response
Respected Sir/Madam
Here I have attached the response to the comments and suggestions of Reviewer 1.

Reviewer 2 Report
Abstract: Important findings of the study needs to be highlighted in the abstract
Introduction
Description of the type of the farmer
Line 33: Be specific on which continents are being discussed
Lines 37-40: Citation must be done
Line 111: Do not write " can't" but cannot in full
Principles of organic farming must be referenced
Results do not address the objects sets
Line 112: The examples must be referenced
Line 128-131: Citation required
Literature Review is not comprehensive, more needs to be done.
There is need to be consistent with one referencing style and not to mix referencing styles
Line 150, Line 161, Line 181, Line 191, Line 198, Line 293, Line 296-303 & Line 344: referencing style needs to be checked
Section on Organic Farming vs Coventional farming must be discussed under literature review section and must come before the principles of organic farming
Study site discussion must be referenced.
Methodology needs to be clearly described illustrating the paradigm and the research design. Methodology section requires citation. There is need for the sample size and detailed description of the participants
Results must be compared with findings from other studies.
Author Response
Respected sir/ Madam
Here I have attached the response to the comments and suggestions of Reviewer 2.

Round 2
Reviewer 1 Report
Thank you to the authors for revising the manuscript. The quality has improved substantially. There are still some issues to work on that I describe below. Also, please get the English language proof-read before the next revision, it is really required.
196-197 I would propose to rephrase: Having briefly established the pros and cons of each mechanism of cultivation…
252-264 Please provide some references in this paragraph
256 Section 2.3 Scope for improving organic agriculture in India is probably not needed, it’s a continuation of 2.2. Organic Farming in India
288 Section 2.4 has the same title as the Section 2.5., and repeats one part of it. Please delete it.
305-321 This section is quite repetitive. I would delete the repetitive parts and merge it with section 2.1.
322-336 Please integrate into section 2.2. Organic Farming in India
337-381 This could be a new subsection Farmers’ suicide in India. In this subsection please also integrate line 207 and onwards till subsection 2.3. and remove all the repetitive parts.
383 Which governmental statistics? Please add a reference
394-406 Please provide references.
406-408 Repetitive and should be deleted
Figure 2 and Figure 3 should be mentioned in the text where it is appropriate.
438-445 Please provide reference and mention Figure 5 in the text where appropriate.
450 Is Section 2.9 really needed? If not integrate it into the previous section where you discuss farmers’ suicide in India and the statistics.
512-520 Are these results or data information? If it’s data, please move the Methodology section.
523 Mention Figure 6 here.
527-548 This also seems like a paragraph that actually belongs to the Methodology section. If it discusses the variables that were used and not the Results, it needs to be moved up to the Methodology section.
568-574 Could you provide a graph here to show the results (if you have them)? I’m wondering how come is it that conventional farmers get higher prices? Shouldn’t the organic farmers actually get a higher price for the organic rice, since organic products have quite clear health benefits? At least this is the case is Europe and the US. If the opposite is true in India, and the market actually rewards conventional producers, this could be discussed in the discussion part. Though I think (if I understood well) that actually in your sample the organic farms get higher profits per acre? In that case, what you wrote earlier about the market rewarding the conventional producers, doesn’t really hold true. If this is the case, please delete this part, it contradicts your results.
592-593 You say H01 is rejected and H02 is accepted. But the in the Table you say the opposite thing in the Results column.
628-629 Please briefly mention these other studies that have shown different or similar results to yours and quote them here.
630 Actually, if I get it right, your results suggest that there is no significant difference between costs of conventional and organic farms, meaning that the argument that organic farming will reduce the costs cannot be supported by your study. This should be explicitly mentioned. You can quote some other studies in India that show different results (organic farming does reduce the costs for farmers), but you need to be explicit about your findings. Please correct me if I misinterpreted your results.
656-663 This part does not really belong to conclusions and is not supported by your results which doesn’t show any difference between the costs of conventional and organic farms. Correct me if I’m wrong. If not, delete this part.
You can, of course, accentuate the higher profitability of organic farming as a way to prevent suicide, since this is supported by your results.
Author Response
Re: Revision of manuscript ID: Sustainability- 455002
Sustainability Journal
Dear Sir/Madam,
Thank you very much for the email which included the reviewers' comments and suggestions on our manuscript entitled "Farmers’ Suicide Versus Organic Farming: A Threat and Opportunity for Sustainable Agricultural Development in India" (Manuscript ID 455002). On behalf of my co-authors, I would like to express my great appreciation to editor and reviewers. We are very encouraged that all reviewers agree with the significance of our findings, and recommend the publication in the Journal of Sustainability. We appreciate their constructive and valuable comments and suggestions; hence, we have revised our manuscript accordingly. Now we would like to submit the revised manuscript for your consideration. Based on the comments and suggestions, we have revised our paper and indicated the changes in the main document.

Reviewer 2 Report
62-69: Reference required
98: Sentence not clear
113: Reference correctly
462: Reference correctly
466: Reference correctly
Result section is not clear. Redo this section. The findings must be clear
Bar Char: No description of the bar chart
Table 5: No description of table.
Discussion: Redo. Clearly show your results and compare with previous studies.
NB: Before the next submission the paper needs to have gone for English language editing and proof reading
Author Response
Response to Reviewers’ Comments – Round 2
Re: Revision of manuscript ID: Sustainability- 455002
Sustainability Journal
Point 1: 62-69: Reference required.
Response 1: Reference has been added.
Point 2: 98: Sentence not clear.
Response 2: Revised the sentence as per suggestion.
Point 3: 113: Reference correctly.
Response 3: Reference was corrected [9].
Point 4: 462: Reference correctly.
Response 4: Reference was corrected [40].
Point 5: 466: Reference correctly.
Response 5: Reference was corrected [41].
Point 6: Result section is not clear. Redo this section. The findings must be clear.
Response 6: As per suggestion, result section has been revised and modified for reader’s clarity. The findings has been modified and clearly explained by data and description. Figure 6 has been newly added and more points added in results section.
Point 7: Bar Chart: No description of the bar chart.
Response 7: Bar Chart description has been added.
Point 8: Table 5: No description of table.
Response 8: Table 5 Description has been added.
Point 9: Discussion: Redo. Clearly show your results and compare with previous studies.
Response 9: According to suggestion, discussion part has been revised and added more points for reader’s clarity. Results had been compared with other studies and discussed in this section.
Point 10: NB: Before the next submission the paper needs to have gone for English language editing and proof reading
Response 10: The authors accepted the comments of reviewer. After received your comments, we have improved the manuscript and made changes accordingly. Following your comments, texts’ errors were checked and corrected by English grammar, and similarly, proof reading was done by native English-speaker.
